# Deletion of *pbpC* Enhances Bacterial Pathogenicity on Tomato by Affecting Biofilm Formation, Exopolysaccharides Production, and Exoenzyme Activities in *Clavibacter michiganensis*

**DOI:** 10.3390/ijms24065324

**Published:** 2023-03-10

**Authors:** Yao Li, Xing Chen, Xiaoli Xu, Chengxuan Yu, Yan Liu, Na Jiang, Jianqiang Li, Laixin Luo

**Affiliations:** 1Department of Plant Pathology, College of Plant Protection, China Agricultural University, Beijing 100193, China; lycau1994@126.com (Y.L.); chenxing2028@163.com (X.C.);; 2Key Laboratory of Surveillance and Management for Plant Quarantine Pests, Ministry of Agriculture and Rural Affairs, Beijing Key Laboratory of Seed Disease Testing and Control, China Agricultural University, Beijing 100193, China; 3Key Laboratory of Integrated Crop Pest Management of Anhui Province, Key Laboratory of Biology and Sustainable Management of Plant Diseases and Pests of Anhui Higher Education Institutes, School of Plant Protection, Anhui Agricultural University, Hefei 230036, China

**Keywords:** *Clavibacter michiganensis*, penicillin-binding proteins, pathogenicity, exoenzymes, exopolysaccharides

## Abstract

Penicillin-binding proteins (PBPs) are considered essential for bacterial peptidoglycan biosynthesis and cell wall assembly. *Clavibacter michiganensis* is a representative Gram-positive bacterial species that causes bacterial canker in tomato. *pbpC* plays a significant role in maintaining cell morphological characteristics and stress responses in *C. michiganensis*. The current study demonstrated that the deletion of *pbpC* commonly enhances bacterial pathogenicity in *C. michiganensis* and revealed the mechanisms through which this occurs. The expression of interrelated virulence genes, including *celA, xysA, xysB*, and *pelA*, were significantly upregulated in △*pbpC* mutants. Compared with those in wild-type strains, exoenzyme activities, the formation of biofilm, and the production of exopolysaccharides (EPS) were significantly increased in △*pbpC* mutants. It is noteworthy that EPS were responsible for the enhancement in bacterial pathogenicity, with the degree of necrotic tomato stem cankers intensifying with the injection of a gradient of EPS from *C. michiganensis*. These findings highlight new insights into the role of *pbpC* affecting bacterial pathogenicity, with an emphasis on EPS, advancing the current understanding of phytopathogenic infection strategies for Gram-positive bacteria.

## 1. Introduction

*Clavibacter michiganensis* (Cm) is the causal agent of tomato bacterial canker and wilt disease, which results in substantial yield losses and catastrophic damage to tomato (*Solanum lycopersicum*) production worldwide. It is a non-motile actinomycete and is recognized as a dominant species that proliferates in the xylem of host plants [1,2]. In particular, it has been confirmed that different *Clavibacter* spp. have high host specificity, as each species is restricted to a narrow host plant species range and survives in plant debris but shows poor survival capabilities in soil environments [3]. However, no Cm-resistant commercial tomato cultivars are available, and Cm disease management principally depends on seed treatment, agricultural practices, and spraying chemicals, such as streptomycin sulfate, copper compounds, and mancozeb [4,5,6].

Infection by *Clavibacter michiganensis* occurs through contaminated seeds, host wounds, and natural openings. Systemic and local symptoms are caused by Cm, which is mediated by particular temperature and humidity conditions [7]. Regarding systemic infection, the proliferation of bacteria in the xylem causes the deterioration of the internal vascular tissues and subsequently culminates in the generation of stem cankers and leaf wilting, which ultimately causes the death of the entire plant [8,9]. When systemic infections occur at an early stage of the host plant lifecycle, such as at the plant seedlings stage, the bacteria extensively spread through the xylem vessels, and the plants show a destructive withering process resulting in yield losses, even when agronomic chemical management is implemented during the infection. In contrast, if the plants are infected during a late developmental stage, their leaves or fruits may appear asymptomatic, exhibit foliar chlorosis, or produce fruits with localized bird’s-eye spots, but the fruit quality and quantity may still be affected [10,11].

Penicillin-binding proteins (PBPs) are pivotal membrane proteins involved in peptidoglycan (PG) biosynthesis, which is responsible for the morphological characteristics of bacterial cells and the mechanical integrity of the cell wall [12,13]. Based on previous studies, bacterial PBPs can be divided into three classifications. The first category includes low-molecular-weight (LMW) PBPs, which are monofunctional proteins that are involved with D, D-carboxypeptidases. The other two families of PBPs are multifunctional enzymes: high-molecular-weight class A PBPs (HMW, aPBPs) polymerize and cross-link with glycans, whereas high-molecular-weight class B PBPs (HMW, bPBPs) exhibit transpeptidase activity. Cell wall peptidoglycan is synthesized by the concerted action of transglycosylation and transpeptidation; as a result, PBPs are essential to bacterial cell elongation and propagation [13,14]. In *Bacillus subtilis*, a lack of aPBPs can affect cell shape: the envelope of such cells is thinner, and the rod-shaped cells are not circumferentially organized [12]. To maintain the integrity of the cell wall, PBP1b, which is a prominent class A PBP, actively renovates *Escherichia coli* cell wall deficiencies, functioning as a cytoskeletal machine [15]. PBP1, encoded by *penC*, mediates transformations to achieve higher levels of chromosomal penicillin resistance in *Neisseria gonorrhoeae* [16]. A total of seven PBPs are contained in *Clavibacter michiganensis*. Additionally, class A HMW PBPC was discovered to be vital for cell morphology in *C. michiganensis*. With the deletion of *pbpC* in Cm, the cell loses its rod shape and becomes intumescent from one pole, thus affecting the synthesis of peptidoglycan and altering bacterial pathogenicity [17]. Moreover, PBPs possess the capacity to covalently bind to bacterial proteins and impede cell division and have drawn much attention as targets of β-lactam antibiotics [18,19,20]. Notably, PBPs have been widely investigated in human pathogens, but fewer studies have been conducted on phytopathogenic bacteria, especially in terms of their role in regulating pathogenicity.

The genomic features of various wild-type *C. michiganensis* strains were sequenced: they contain a 3.3–3.6 Mbp chromosome and generally two plasmids, pCM1 (31–59 kb) and pCM2 (64–109 kb) [21]. The molecular underpinnings of *C. michiganensis* indicated that a chromosomal *chp/tomA* pathogenicity island (PAI), which encodes serine proteases and multiple other proteases, is responsible for pathogenicity. Meanwhile, *tomA* encodes tomatinase, which is involved in the detoxification of α-tomatine [22]. In addition, the majority of tomato-pathogenic *C. michiganensis* strains harbor the two plasmids. The virulence gene *celA*, encoding β-1,4-endocellulase, is located on pCM1 and has been reported as an essential pathogenicity factor. The second factor contributing to virulence is *pat-1*, which encodes a putative serine protease and has been identified on plasmid pCM2 [23,24]. A number of transcriptional regulators also associated with virulence, such as *Vatr1* and *Vatr2*, are located in chromosomal regions [25].

A bacterial biofilm is a rigid surface matrix composed of proteins, DNA oligomers, exopolysaccharides, and peptides, which act as protective shields against pathogens [26,27]. *Corynebacterium pseudotuberculosis* produces a biofilm matrix that can protect it against antibiotics, environmental stresses, and host immune responses [28,29]. This strategy provides an advantage for biofilm-forming strains in colonizing and surviving in host tissue compared with non-biofilm-forming strains [30]. *Pseudomonas aeruginosa* biofilms are embedded within extracellular polymeric substances and are one of the most crucial virulence factors, causing destructive chronic and acute infections in humans [31,32]. In plant pathogenic bacteria, biofilm formation capacity enhances bacterial stress tolerance and facilitates the colonization of plant xylem and seeds [33]. *Clavibacter nebraskensis*, which causes bacterial Goss’s wilt of corn, has highly virulent strains that show an aggregation phenotype within the xylem; the moderately virulent strains form biofilms around the xylem [34]. Likewise, the bacterial exopolysaccharides (EPS) are implicated in virulence and act as adhesion factors in *Burkholderia* species, which are typically mammalian pathogens, using the same strategies as bacterial flagella for transmission and infection [35]. In plant pathogenic bacteria, *Erwinia amylovora*, the pathogen causing fire blight in Rosaceae, produces EPS, such as amylovoran, levan, and cellulose. Amylovoran is associated with the physical blockage of plant vascular tissues [36,37]. Moreover, amylovoran is necessary for *E. amylovora* biofilm formation and contributes to bacterial colonization and migration in xylem vessels [38,39]. Most of the well-characterized EPS and their function in pathogenicity were studied in species of *Xanthomonas*, *Pseudomonas*, *Pantoea*, and *Xylella* [40]. Nevertheless, the role of EPS in the pathogenicity of Gram-positive bacteria, especially in *C. michiganensis*, has not yet been well studied.

Indeed, some unknown factors in *C. michiganensis* may affect the disease development and the characteristics of symptoms in the fields [10]. In this study, the mechanism of enhanced virulence of the *pbpC* deletion mutants was systematically studied through the production of biofilm and EPS, as well as exoenzyme activities. Ultimately, we hypothesized that EPS and cell wall peptidoglycan cross-linkage act as pivotal elements contributing to the modulation of pathogenicity, which provides a novel perspective for understanding the mechanism of pathogenicity in *C. michiganensis*.

## 2. Results

### 2.1. Expression Level of Pathogenesis-Related Genes in C. michiganensis pbpC Mutants

A series of *pbpC* deletion mutant, complementation, and *pbpC* over-expression strains were successfully generated in two *C. michiganensis* strains (BT0505 and GS12102). The expression level of *pbpC* could not be detected in GS12102-△*pbpC* or BT0505-△*pbpC* mutants. Compared with corresponding wild-type strains, the expression levels of *pbpC* were significantly higher in the complementation and over-expression strains (Appendix A).

Three housekeeping genes, *gyrB*, *gapA*, and *bipA*, were employed as reference genes for the normalization of the relative expression of nine pathogenic-related genes in *C. michiganensis*. The expression levels based on qRT-PCR were measured after all the bacterial strains had been incubated in LB medium supplemented with tomato seedling grinding homogenate (10 mL/L) for 20 h. Based on the previous annotated genome study, *CelA* is located on the plasmid pCM1 and encodes a cellulase, whereas *CelB* is located on the chromosome. The relative expression of *CelA* was 4- to 5-fold higher in △*pbpC* mutants of BT0505 and GS12102 compared with that in wild-type and other transformants (Figure 1A). The expression of *CelB* was slightly increased compared with that of the wild-type strains (Figure 1B). On the other hand, we examined the relative expression of *pat-1*, which encodes a serine protease located on plasmid pCM2. As expected, strain GS12102 and its derivatives showed no distinct changes in the relative expression of *pat-1*, and the expression of *pat-1* could not be detected in strain BT0505 and its derivatives because of their lack of pCM2 (Appendix A).

Various cell-wall-degrading enzymes and genes contribute to *C. michiganensis* infection of tomato. In the current experiment, three genes were examined for further investigation: xylanases encoded by *xysA* and *xysB* and pectate lyase encoded by *pelA*, which are putatively involved in canker production and tissue maceration. The relative expression levels of these three genes showed the same tendency, with the transcription levels all higher in the △*pbpC* mutants than in the wild-type strains and other transformants for both strains, BT0505 and GS12102. The relative expression of *xysA* and *xysB* exhibited statistically significant 6–8- and 5–6-fold increases (Figure 1C,D), respectively. A 3–4-fold increase in the expression level of *pelA* was also induced (Figure 1E).

It is widely considered that the serine protease family is an important pathogenic factor in *C. michiganensis* [41]. As a result, it is very meaningful to determine the expression levels of the chromosome genes *chpC* and *ppaA*, which encode serine protease family S1 (chymotrypsin) and extracellular serine protease, respectively, in △*pbpC* mutants. The relative expression levels of these two genes showed no significant differences (Figure 1F,G). The results indicated that cell-wall-degrading enzymes were significantly increased in *pbpC* deletion mutants in *C. michiganensis*, which might be related to *pbpC* being responsible for peptidoglycan biosynthesis and cross-linking. However, we observed no noticeable changes in the relative expression of serine protease, which assists in bacterial colonization and virulence.

### 2.2. Endocellulase and Amylase Secreted by pbpC Derivatives

Bacterial cell-wall-degrading enzymes such as cellulase, amylase, xylanase, and polygalacturonase may be pivotal virulence factors that cause wilting in tomatoes by degrading plant cell wall components [42]. In this study, the secretion of endocellulase activity was qualitatively assessed through plate assays using Congo red staining. Hydrolysis halos surrounding the bacterial colonies were observed in visualization, indicating a notable enhancement in exoenzyme activities. Compared with the wild-type and complementation strain, the △*pbpC* mutant had stronger cellulase activities through incubation with the substrate carboxymethylcellulose (CMC). The ratio of the diameter between bacterial colonies and the clear halo zone was significantly different (*p* < 0.05) in cellulose hydrolyzation, with clear halo zone values of 1.04 ± 0.08 and 1.07 ± 0.03 cm in the wild-type BT0505 and GS12102 strains, respectively; 1.34 ± 0.07 and 1.31 ± 0.07 cm in the △*pbpC* mutants derived from BT0505 and GS12102, respectively; and 1.10 ± 0.05 and 1.11 ± 0.06 cm in the *pbpC*-comp strains, respectively (Figure 2A). The amylase activity of △*pbpC* mutants was extremely activated, which was reflected by the hydrolytic halo zone. It was observed from the violet background staining, which showed diameters of 1.75 ± 0.04 and 1.93 ± 0.05 cm in wild-type BT0505 and GS12102 strains, respectively; 3.02 ± 0.08 and 3.27 ± 0.03 cm in the △*pbpC* mutants, respectively; and 1.77 ± 0.05 and 1.74 ± 0.08 cm in the *pbpC*-comp strains, respectively (Figure 2B). Consequently, the secretion of exoenzymes was largely enhanced by a lack of *pbpC* in both strains BT0505 and GS12102 of *C. michiganensis,* which could be a factor modulating bacterial pathogenicity.

### 2.3. Biofilm Formation of pbpC Derivatives

The results of the biofilm assays indicated that all the *C. michiganensis* strains had taken the shape of clear biofilm fringes on the wall surfaces of the polystyrene tubes. However, the mutants that lacked *pbpC* exhibited stronger biofilm formation than the wild-type strains and their derivatives (Figure 3A). Quantification assays were performed by staining with crystal violet and washing with ethanol, then the optical density of the solution was measured at 590 nm (OD_590_). The results exhibited that the *pbpC* deletion mutants formed significantly (*p* < 0.05) more biofilm than the wild-type and *pbpC*-comp strains, both in BT0505 and GS12102 (Figure 3B). This finding revealed that *pbpC* has effects on biofilm formation in *C. michiganensis*.

### 2.4. EPS Isolation of pbpC Derivatives and Virulence Testing In Vitro

The EPS from liquid-shaking cultures produced by different *C. michiganensis* strains were extracted and isolated successfully. Bacterial cells were removed by centrifugation after ethanol precipitation overnight at 4 °C. The EPS production without proteins was obtained and weighed after lyophilization. The quantitative results demonstrated that the mass of EPS isolated from △*pbpC* mutants was significantly higher than those of wild-type, *pbpC*-comp, and *pbpC*-OE strains (Figure 4). The EPS produced by the △*pbpC* mutants was 2 g/L of bacterial cells (10^10^ CFU/mL).

A virulence test by the foliar spraying of a gradient dilution series of EPS solution was performed to determine whether EPS elicited an immune response in host plants. In this experiment, EPS solution was diluted to concentrations of 0.01, 0.1, 1, 2, and 5 mg/mL, which was sprayed onto fresh tomato leaves in a Petri dish with moist sterile gauze at 4 °C. As the daily spraying proceeded for 7 days, neither the treatments nor the negative controls showed conspicuous disease symptoms or hypersensitive reactions (Appendix A). Thus, it was hypothesized that tomato leaf tissues were stable enough and hardly infiltrated into the epithelial cells by EPS compounds in vitro.

### 2.5. Effect of EPS on Tomato and Tobacco Leaves

EPS, including capsular polysaccharides, play a crucial role in bacterial adhesion and provide a protective barrier around cells [33]. *N. benthamiana* and *S. lycopersicum* were employed to assess whether EPS-induced confluent necrosis, a typical hypersensitive reaction (HR), occurred in the infiltrated areas of host plants. We observed that the tobacco leaves showed necrotic lesions after infiltration by EPS (Figure 5A). After decolorization by shaking with ethanol, the necrotic lesions were much more evident compared with those in the water control and sucrose solution treatment. No significant difference was observed for 2 and 5 mg/mL EPS treatments on tobacco leaves. When tomato leaves were infiltrated with a gradient of EPS solution (0.5–5 mg/mL), the necrotic lesions significantly differed in a dose-dependent way (Figure 6). High doses of EPS from 1 mg/mL upward contributed to necrosis; however, low doses of 0.5 mg/mL EPS induced no significant changes compared with the negative control.

The tomato stem cracking symptoms and canker area affected by EPS were assessed at 14 dpi. The lesions caused by the GS12102-△*pbpC* mutants were much larger than those caused by the wild-type, *pbpC*-comp, and *pbpC*-OE GS12102 strains; the canker areas were 28.46 ± 1.01, 17.47 ± 1.25, 18.24 ± 1.55, and 16.75 ± 1.50 mm^2^, respectively (Figure 7A,C). It was found that the enhancement in pathogenicity induced by △*pbpC* in *C. michiganensis* was a consistent result, as we also observed in our previous study on strain BT0505 [17]. Furthermore, bacterial EPS was also supported as an influencing factor for pathogenicity by the assay of injecting EPS into wounds after puncturing, which led to lesions of 10.53~±~1.53 mm^2^ on tomato stems (Figure 7B,C). Moreover, with the injection of EPS after *C. michiganensis* inoculation, the canker lesion sizes were significantly (*p* < 0.05) larger than those of the inoculated plants without EPS injections. The canker areas caused by the *pbpC* mutants, wild-type, *pbpC*-comp, and *pbpC*-OE strains were 34.62 ± 1.63, 23.60 ± 0.80, 26.31 ± 1.95, and 23.62 ± 1.29 mm^2^, respectively (Figure 7B,C). However, the number of bacterial cells reisolated from the lesions did not significantly differ among treatments (Figure 7D). The results revealed that EPS is a critical factor for increasing pathogenicity in *C. michiganensis*.

## 3. Discussion

*C. michiganensis* is a quarantine pathogen that causes the devastating bacterial canker of tomato in Europe, Africa, the Americas, and Asia. Understanding the survival strategies and pathogenic mechanisms of *C. michiganensis* is crucial for disease management. It was revealed that the plasmid genes (including *pat-1*, *ppaJ*, *CelA*, *phpA*, and *phpB*), plasmid quantity, PAI on the chromosome, and transcription regulatory factors are involved in the pathogenicity of *C. michiganensis* [43,44,45]. *ChpG* encodes a transmembrane protein that mediates the immune response of multiple eggplant varieties, whereas some virulence factors remain undiscovered and unevaluated [25]. The high relative expression levels of *CelA*, *xysA*, and *xysB*, which encode cellulase or xylanase, in the △*pbpC* mutants indicated that the canker symptom is primarily caused by cell-wall-degrading enzymes. These enzymes function in the proteolysis of specific plant substrates and ultimately result in host plant cell death. In contrast, the relative expression levels of *ChpC*, *ppaA*, and *pat-1*, which are associated with the serine proteases family, showed no significant difference among the △*pbpC* mutants, wild-type, and complementation strains. This provides an explanation of the equal level of bacterial populations reisolated from the inoculated tomato seedlings [44,46].

The bacterial cell wall peptidoglycan layer contributes to cell shape and rigidity and plays a vital role in growth and survival. The class A PBPs are essential for peptidoglycan assembly at the terminal stage of cell wall synthesis [47]. Previously, we found that *C. michiganensis* possesses seven different candidate PBPs, namely PBPA, PBPB1, PBPC, PBPD, PBBE, PBPX, and DacB. The first two are class B HMW PBPs, which encode monofunctional transpeptidases, whereas PBPC and PBPD are designated class A HMW PBPs, which catalyze transglycosylation and transpeptidation as bifunctional enzymes. The other three, PBPE, PBPX, and DacB, are LMW PBPs. All of the seven PBPs have similar structures to their homologous proteins in *E. coli*. Deletion of single and double genes of PBPs in *C. michiganensis* demonstrated that partial PBPs exhibit functional redundancy [13]. Notably, the △*pbpC* mutants exhibited a range of phenotypic variations compared with their parental strains and their derivatives. Regarding cell morphology, the △*pbpC* mutants were not rod-shaped but instead had a thinner cell wall, as observed by transmission electron microscopy and atomic force microscopy. The △*pbpC* mutants of *C. michiganensis* strain BT0505 were more sensitive to lysozymes and environmental stress response, such as ultrasonication and Cu^2+^ treatment [17,48]. However, the mechanism of these phenotypic changes in the △*pbpC* mutants associated with their increased virulence in tomato seedlings was firstly demonstrated in this study.

Two strains of *C. michiganensis* GS12102 and BT0505, which differ in their geographic origin and plasmid number, were used to prepare △*pbpC* mutants and a series of derived strains. It was revealed that the thickness of the cell wall peptidoglycan layer of strain BT0505 △*pbpC* mutants is thinner and the number of cross-links is decreased compared with those of the wild-type strain [48]. Accordingly, a lack of *pbpC* in *C. michiganensis* appeared to result in a more fragile cell wall. Thus, we hypothesize that the secretion of cell-wall-degrading enzymes and serine proteases may be affected by the delicate cell wall of the △*pbpC* mutants. The results of the exoenzyme assays demonstrated that both cellulase and amylase activities intensified in the △*pbpC* mutants compared with those of the other derivative strains. Cellulase activity and secretion efficiency can promote the bacterial invasion of host plants [42]. Our results partially demonstrate the increase in pathogenicity as being caused by the absence of PBPC in *C. michiganensis*.

In natural cases, microbes can produce three kinds of polysaccharides: intracellular storage polysaccharides (glycogen), capsular polysaccharides (linked to the microbial surface), and extracellular polysaccharides (crucial for bacterial pathogenicity and biofilm formation) [49,50]. Barber et al. (1997) indicated that the exopolysaccharide production and virulence of *Xanthomonas* spp. are mediated by an inimitably evolved two-component regulatory system as diffusible signal factors (DSFs) [51]. Bacterial biofilms, which are composed of multiple extracellular polysaccharide substances, can be used as a shield for the encapsulated bacteria as protection against adverse conditions. Numerous studies have confirmed that EPS function as vital factors associated with biofilm formation both in human pathogenic bacteria and phytopathogens [33,38,49,50,52]. In this study, the formation of biofilm in the △*pbpC* mutants was significantly increased compared with that of the wild-type, complementation, and *pbpC* over-expression strains, which may play a primary role in the increased EPS production in the △*pbpC* mutants [49]. In the future, researchers should explore which genes are involved in EPS biosynthesis and transport in *C. michiganensis* and whether these genes are upregulated in △*pbpC* mutants.

In previous studies of *C. michiganensis*, the composition of EPS was successfully detected by high-performance liquid chromatography and proton nuclear magnetic resonance spectroscopy. Its approximate molar ratio was 4:2:2:2:1:3 for l-fucose, d-galactose, d-glucose, pyruvate, succinate, and acetate, respectively. The composition and yield of EPS isolated from the *C. michiganensis*-infected plant tissues were similar to those of EPS isolated from cultured bacterial cells [53]. However, Bermpohl et al. (1996) identified the plasmid-free strain CMM100, a derivative of the standard strain NCPPB382, which forfeits pathogenicity but produces EPS in quantities identical to those of the parental strains, and proposed that EPS may not determine virulence but instead synergistically intensify wilting symptoms [54,55,56]. *Clavibacter* species produce distinct two types of EPS: acidic polymer and neutral EPS with low molecular weight [57]. The EPS from *Clavibacter sepedonicus* and *Clavibacter insidiosus* can destroy plant membranes and be used to differentiate susceptible and resistant varieties [53,58,59]. As for *C. michiganensis*, the EPS isolated from the strains used in our study induced a hypersensitive reaction in both *S. lycopersicum* and *N. benthamiana* and exhibited a dose-dependent effect in vivo. The EPS concentration of 1 mg/mL or more contributed to leaf necrosis by infiltration assay, which was equivalent to the amount of EPS secreted by 10^9^–10^10^ CFU cells of *C. michiganensis*. The bacterial titer of *C. michiganensis* in inoculated tomato stems can reach 10^9^ CFU/g, which is sufficient to produce enough EPS to interact with the host plant. Given the enhanced pathogenicity of the △*pbpC* mutants, EPS may be responsible for increasing the stem canker areas. When injecting EPS into the puncturing wounds, tomato stems showed more obvious canker symptoms than mock controls. As a result, EPS may conglutinate together and block the plant stem xylem vascular system, resulting in restricted water movement [57]. A higher-capacity biofilm formation and increased EPS yields induced stronger resistance of the △*pbpC* mutants to the host plant defense response. The chemical composition of EPS from different *C. michiganensis* strains and PBP mutants also merits further investigation to confirm whether variants of this component can alter pathogenicity.

Bacterial cell wall peptidoglycan, as an emblematic pathogen-associated molecular pattern (PAMP), can be recognized by the plant immune reaction [60,61]. The thinner cell wall of the △*pbpC* mutants, with a distorted peptidoglycan layer, led to the reduction in peptidoglycan cross-links. The structural variation in the *pbpC* mutants resembling a “ridge and groove” surface, compared with the relatively standard flat surface in wild-type and complementation strains, as observed by atomic force microscopy, may have affected host plant recognition [62]. Nevertheless, the reduction in cell wall peptidoglycan in the △*pbpC* mutants (data not shown) may also have led to the interference of host plant recognition. In *Helicobacter pylori,* it forms a special muropeptide based on the structural variation in bacterial peptidoglycan, which helps the pathogen to bypass the host immune system and respond to environmental stress [63]. Thus, further studies should concentrate on whether alterations in the bacterial peptidoglycan surface structure and content may increase pathogenicity in the △*pbpC* mutants.

Overall, our findings indicate that a lack of *pbpC* in *C. michiganensis* increased pathogenicity in tomato by the secretion of more extracellular cell-wall-degrading enzymes and an increase in the production of bacterial biofilm and EPS. As the cross-links in the cell wall peptidoglycan layer decrease, a hypothesis regarding the interaction between the *C. michiganensis* cell wall with plant immune recognition is required [64]. The original findings reveal a future understanding of bacterial pathogenicity and provide a possible idea for controlling bacterial diseases by inhibiting the biosynthesis of EPS and other potential virulence factors [50,65,66]. However, more details are needed to support the present hypothesis that the chemical composition of EPS contributes to plant–microbe interactions and the establishment of disease symptoms caused by phytopathogens, particularly Gram-positive bacteria.

## 4. Materials and Methods

### 4.1. Strains, Plasmids, and Growth Conditions

The bacterial strains involved in this study are detailed in Appendix A. Two wild-type strains of *C. michiganensis* were used in this study. Strain GS12102 harbors two plasmids, pCM1 and pCM2; the other strain, BT0505 only harbors pCM1. The knock-out mutants of *pbpC* and the complementation strains were prepared as previously described [17,67]. The *C. michiganensis* strains were grown on Luria–Bertani (LB) agar or in LB broth medium (10 g of tryptone, 5 g of yeast extract, 5 g of NaCl, and 16 g of agar for solid medium per liter) at 28 °C for 72 h, whereas the *Escherichia coli* strains were cultured at 37 °C with shaking as required (at 160 or 200 rpm). For the mutant strains, 10 mg/L of chloramphenicol and 50 mg/L neomycin were added when selection was required by appropriate antibiotics.

### 4.2. Construction of pbpC Over-Expression Strains

The *pbpC* over-expression strains were generated from the parental strains GS12102 and BT0505 by using the shuttle vector pHN216 [68]. Total genome DNA of *C. michiganensis* was extracted with a TIANamp Bacterial DNA Kit (TIANGEN Biotech, Beijing, China) according to the manufacturer’s instructions. The full-length sequences of *pbpC*, including the constitutive promoter J23119 (BBa_J23119 registered in standard biological parts; www.partsregistry.org) were amplified by PCR, which we cloned into the vector between the *Hind*III/*EcoRV* restriction sites and transformed into *E. coli* strain DH5α. The over-expression strains were obtained by electroporation as described in a previous study [69]. Ultimately, the transformants were verified by DNA sequencing and qPCR assays.

### 4.3. RNA Extraction and cDNA Synthesis

To prepare total RNA from in vitro-cultured bacteria, wild-type strains and their derivatives were grown in liquid LB medium supplemented with 10 mL/L filter-sterilized tomato stem homogenate, which was intended to simulate the interaction between *C. michiganensis* and host tomato plants during propagation. The tomato stems were collected and ground into homogenate in liquid nitrogen, which was filtered with sterilized gauze to remove fibers, and then we collected the supernatant after centrifugation. The filtered stem homogenate was introduced into the liquid medium for Cm culture [22]. Bacterial cells with a final OD_600_ of 1.0 were prepared and suspended in nuclease-free sterile water. Bacterial suspensions were harvested by centrifugation, and the cell pellets were subsequently frozen in liquid nitrogen prior to RNA isolation. To promote cell lysis, freshly prepared Lysozyme solution was added to TE buffer, which was then resuspended and ultimately incubated at 37 °C for 10 min. An SV Total RNA Isolation System Kit (Promega Corporation, Madison, WI, USA) was used to extract bacterial RNA according to the manufacturer’s instructions.

The isolated RNA was treated with gDNA Eraser (Takara Bio, Kusatsu, Japan) to remove the residual genomic DNA, which was validated by quantitative reverse-transcriptase (qRT)-PCR before reverse transcription. The synthesis of the cDNA was conducted using 1 μg of total RNA with a PrimeScript RT Reagent Kit (Takara Bio, Kusatsu, Japan). The cDNA was diluted and then stored at −20 °C until qRT-PCR assays.

### 4.4. Gene Expression Analysis by qRT-PCR

The pathogenicity genes and reference genes with primers based on previous studies are listed in Appendix A [23,70]. The qRT-PCR was performed by using a 7500 Real-time PCR System (Applied Biosystems, Waltham, MA, USA) following the protocol of the SYBR Premix DimerEraser Kit with Dye (Takara Bio, Kusatsu, Japan). The 4-fold diluted cDNA templates were amplified in triplicate according to the following protocol: 95 °C at 30 s for initial denaturation; 40 cycles of 95 °C for 10 s; 58 °C for 30 s; and 72 °C for 20 s. The relative quantification was performed based on the cycle threshold (C_t_) value and calculated by using the 2^−△△Ct^ method. All experiments were conducted in triplicate.

### 4.5. Exoenzyme Activity Measurement Assays

Endocellulase and amylase activities were evaluated in this experiment. M9 minimal medium (16 g agar, 0.492 g of MgSO_4_·7H_2_O, 0.014 g of CaCl_2_·2H_2_O, 7 g of Na_2_HPO_4_·7H_2_O, 3 g of KH_2_PO_4_, 0.5 g of NaCl, 1 g of NH_4_Cl and 5 g glucose, 0.1% *w*/*v* yeast extract as carbon source per liter) containing 0.5% (*w*/*v*) carboxymethyl cellulose (CMC, Sigma-Aldrich, St Louis, MO, USA) was used to assay the cellulolytic activity; 0.05% (*w*/*v*) soluble starch was used to assay amylase activity. Bacterial suspensions (OD_600_ = 0.3) of 10 µL aliquots were dripped onto M9 medium and were incubated at 28 °C for 5 d. Thereafter, the plates were stained with 0.1% Congo red (Sigma-Aldrich, St Louis, MO, USA) for 2 h and I_2_-KI (53.12 g KI, 0.203 g solid I_2_ per 100 mL) for 15 min, separately. Stained plates were bleached with 1 M NaCl for red plates and 70% (*v/v*) ethanol for violet plates, repeatedly. Exoenzyme activities were indicated by the formation of clear yellowish halo zones around bacterial colonies. All measurements were estimated and calculated based on the diameters of the clear halo and bacterial colonies.

### 4.6. Biofilm Measurement Assays

Biofilm measurement assays were performed as previously described with a few modifications [52]. The *C. michiganensis* strains were activated and cultured at 28 °C and 160 rpm for 20 h in LB liquid medium, which inducted post-log-phase cells at 10^8^ CFU/mL. Then, 1 mL of each bacterial culture was transferred into a polystyrene culture tube containing 9 mL of liquid medium, which was then incubated at 28 °C for 5 d without shaking. The cultures were pipetted and then rinsed with sterile water. After fixation in a drying oven at 80 °C for 20 min, the biofilms on the wall surfaces of the tube were stained with 0.1% crystal violet for 30 min and were then washed with distilled water. The tubes were recorded with a camera (Canon 6D2, Tokyo, Japan). The biofilms were analyzed by visualization and quantitation by solubilizing the stained biofilms with absolute ethanol for 2 h followed by the OD_590_ measurement of the suspension with a spectrophotometer.

### 4.7. Isolation and Quantification of Bacterial EPS

For EPS extraction and isolation, bacteria were grown in LB liquid medium for 20 h at 28 °C with shaking at 160 rpm. Thereafter, all of the liquid cultures were transferred into a new flask with 50 mL of LB liquid medium and were incubated under identical conditions for 48 h to enrich the culture for the bacterial cells. After removing bacterial cells by centrifugation at 10,000 rpm for 15 min, the resulting supernatant was added to three volumes of ethanol and stored overnight at 4 °C. The suspension was subjected to cryogenic centrifugation, and the sediments were dissolved in sterile water. Proteins and insoluble substances were removed as previously described, with some modifications [56,71]. The proteins were denatured by adding an equal volume of Saveg reagent (trichloromethane: n-butyl alcohol = 5:1, *v:v*), and the mixtures were shaken thoroughly for 20 min. After removing the organic phase, the aqueous phase was subjected to centrifugation at 10,000 rpm for 15 min. The extraction was performed three times, with the aqueous phase removed each time. The EPS precipitate was weighed after lyophilization and stored at 4 °C before use.

### 4.8. Plant Material and Growth Conditions

Tobacco (*Nicotiana. Benthamiana*) seeds were germinated in a pot filled with nutrient soil and vermiculite at a ratio of 5:1 (*w/w*). After two weeks, the seedings were transplanted into new pots filled with an identical substrate, with one plant per pot. The plants were grown in a growth chamber with an 18/6 h light/dark cycle at 24 ± 2 °C with a relative humidity of about 70%. Three-week-old plants at the four-leaf stage were used for the HR test [72].

Tomato (Solanum lycopersicum cv. MoneyMaker) seeds were grown in several pots containing a mix of turfy soil, vermiculite, and poultry manure in a ratio of 5:5:1 (*w*/*w*/*w*). Plants were cultivated with a 14 h-light and 10 h-dark photoperiod in a greenhouse at 18–28 °C. Six-week-old tomato plants, at the 6–8-leaf stage, were used for virulence testing.

### 4.9. Virulence and HR Test

For the HR test, EPS solution was diluted by sterile water into a series with concentrations of 0.01, 0.1, 1, 2, and 5 mg/mL, and infiltrated into leaves of *N. benthamiana* and *S. lycopersicum* in an equivalent volume of 0.5 mL with a disposable sterilized syringe, respectively. Sterile water and 5 mg/mL sucrose solution were employed as the mock and negative control treatments.

The stem-pricking inoculation method was performed in this study. All of the tested *C. michiganensis* strains were cultured at 28 °C for 3 days on LB medium. The tomato stems were pricked with a sterilized puncture needle. Then, one single colony was picked and penetrated into the wound made between the two cotyledons. Furthermore, we injected EPS solution isolated from wild-type, △*pbpC* mutant, *pbpC*-comp, and *pbpC*-OE strains, separately, at a moderate concentration of 1 mg/mL, into the wounds 1 d after inoculation. For the mock control, inoculation was performed with sterile LB agar or only EPS solution of an approximately similar volume after pricking. The canker symptoms, which were recorded and evaluated at 14 d post-inoculation (dpi), and the lesion areas were calculated using Fiji software (ImageJ, version 1.53c, Bethesda, MD, USA) as previously described [67]. All tests were performed three times with six replicates per treatment (n = 6). Both virulence and HR testing were repeated at least three times.

### 4.10. Bacterial Population Measurement and Validation in Planta

Bacterial populations were measured at 14 dpi. Inoculated plants were harvested to obtain stem segments from the pricked site that were approximately 1 cm in length and 0.5 g in weight. The stem segments were frozen with liquid nitrogen and then ground using a ball-milling machine. The tissues were serially diluted 10-fold with 0.85% (*w*/*v*) NaCl solution and then were plated on LB medium and incubated at 28 °C for 3 d. The bacterial populations are expressed as CFU per gram of fresh weight of stem segments.

### 4.11. Statistical Analysis

All experiments were repeated three times, and data are expressed as means ± standard errors. The visualization of the datasets and statistical analysis was conducted using GraphPad Prism 7.0 software (GraphPad Prism Software, San Diego, CA, USA). The significance of the data was performed by a nonparametric Mann–Whitney U-test (*p* < 0.05) and one-way analysis of variance (ANOVA) to compare differences between groups. For the comparisons between groups, following ANOVA, Duncan’s multiple range test (*p* < 0.05) was used.

## Figures and Tables

**Figure 1 ijms-24-05324-f001:**
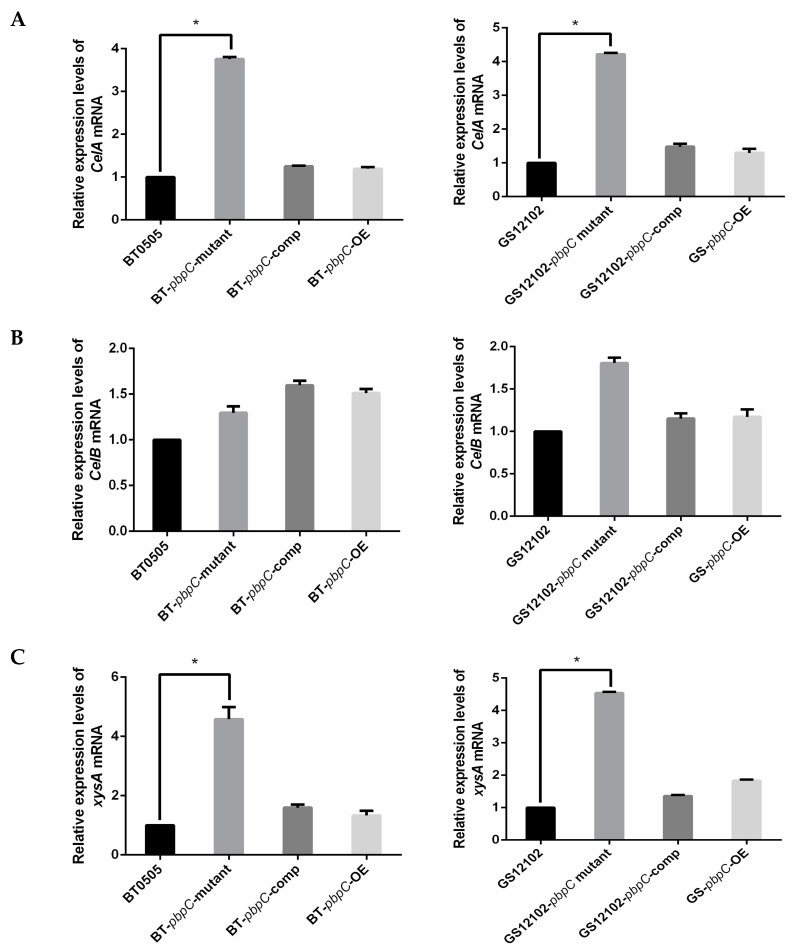
Relative expression levels of (**A**) *CelA*, (**B**) *CelB*, (**C**) *xysA*, (**D**) *xysB*, (**E**) *pelA* (**F**) *ChpC*, and (**G**) *ppaA* determined by quantitative reverse-transcription polymerase chain reaction (qRT-PCR) for BT0505 and GS12102, the △*pbpC* mutants, *pbpC*-comp, and *pbpC*-OE strains compared with their wild-type strains. Relative gene expressions were normalized by *gyrB*, *gapA*, and *bipA*, which we employed as housekeeping references. Results are presented as means of three independent experiments with standard deviation (n = 3). Asterisks (*) indicate significant differences (*p* < 0.05).

**Figure 2 ijms-24-05324-f002:**
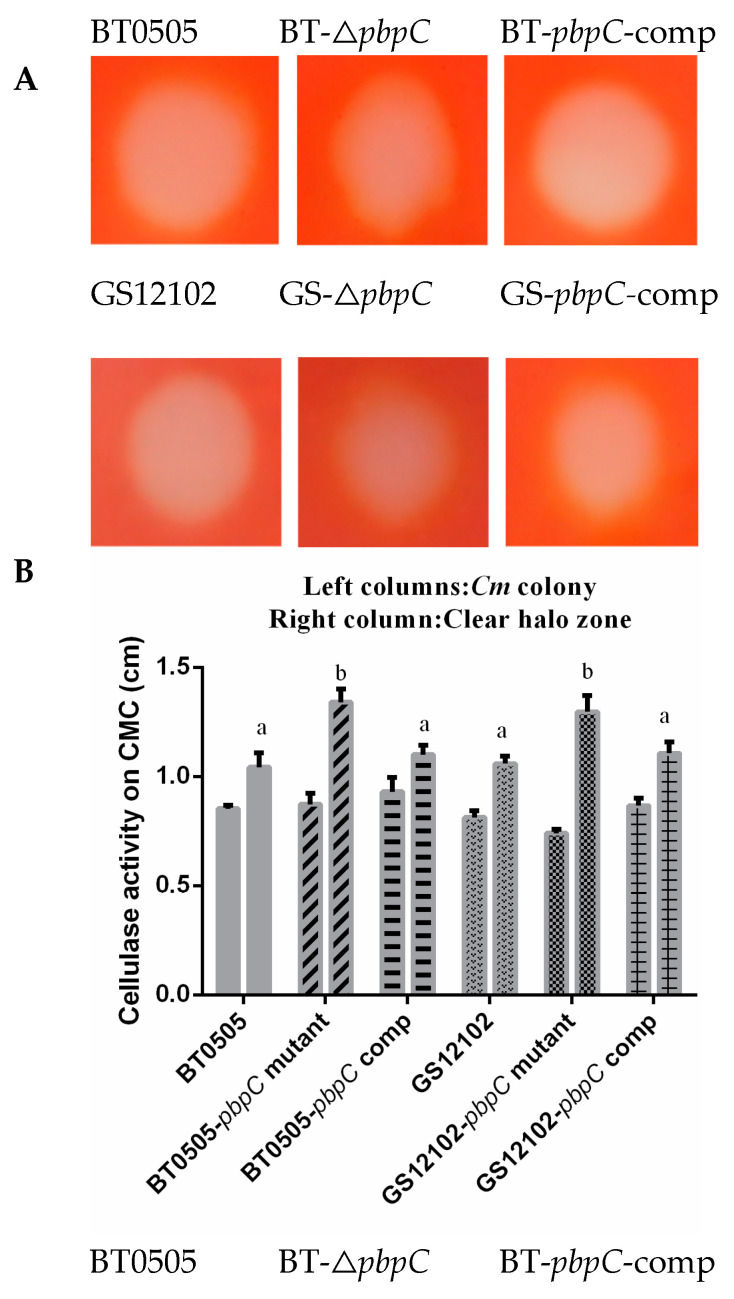
Endocellulase and amylase activities produced and secreted from wild-type, △*pbpC* mutants, and *pbpC*-comp of strains *Clavibacter michiganensis* GS12102 and BT0505. (**A**) Endocellulase activity was assayed on M9-carboxymethyl cellulose (CMC) plates. Plates were stained with 0.1% Congo red, washed with 1 M NaCl, and then photographed. (**B**) Diameter and clear halo zone of bacterial colonies were evaluated for endocellulase activity assay. (**C**) Amylase activity was assayed on soluble starch plates. Plates were stained with I_2_-KI solution, washed with ethanol, and then photographed. (**D**) Results of bacterial colonies diameters and clear halo zones for amylase activity assay. Endocellulase and amylase activities were assessed by clear halo zones around bacterial colonies. Error bars represent standard error (n = 6). Different letters represent statistically significant differences by Duncan’s multiple range test (*p* < 0.05).

**Figure 3 ijms-24-05324-f003:**
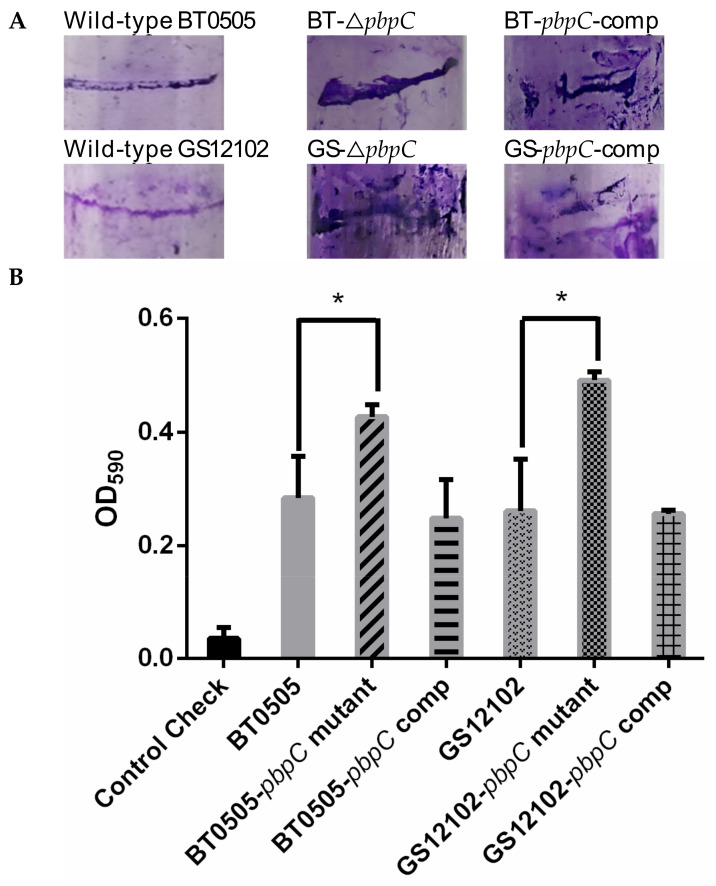
Evaluation of biofilm formation for wild-type, △*pbpC* mutants, and *pbpC*-comp of *C. michiganensis* strains GS12102 and BT0505. (**A**) Images of biofilm formed in polystyrene culture tubes characterized by stained crystal violet fringes around the tube wall. (**B**) Biofilm quantification followed with ethanol washing of crystal violet and spectrophotometer measurements at optical density 590 (OD = 590 nm). Data are presented as means with standard error. Asterisks (*) indicate significant differences between wild-type strains and △*pbpC* mutants as calculated by ANOVA (*p* < 0.05, n = 6). Error bars indicate standard deviation (SD).

**Figure 4 ijms-24-05324-f004:**
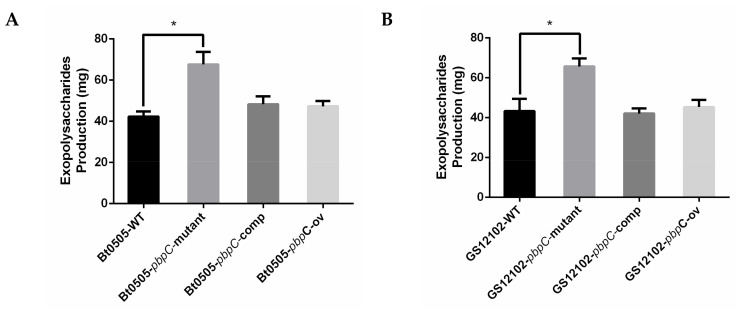
EPS production of wild-type, △*pbpC* mutants, *pbpC*-comp, and *pbpC*-OE of *Clavibacter michiganensis* strains (**A**) BT0505 and (**B**) GS12102. Error bars represent standard error (n = 6). Asterisks (*) indicate significant differences (*p* < 0.05).

**Figure 5 ijms-24-05324-f005:**
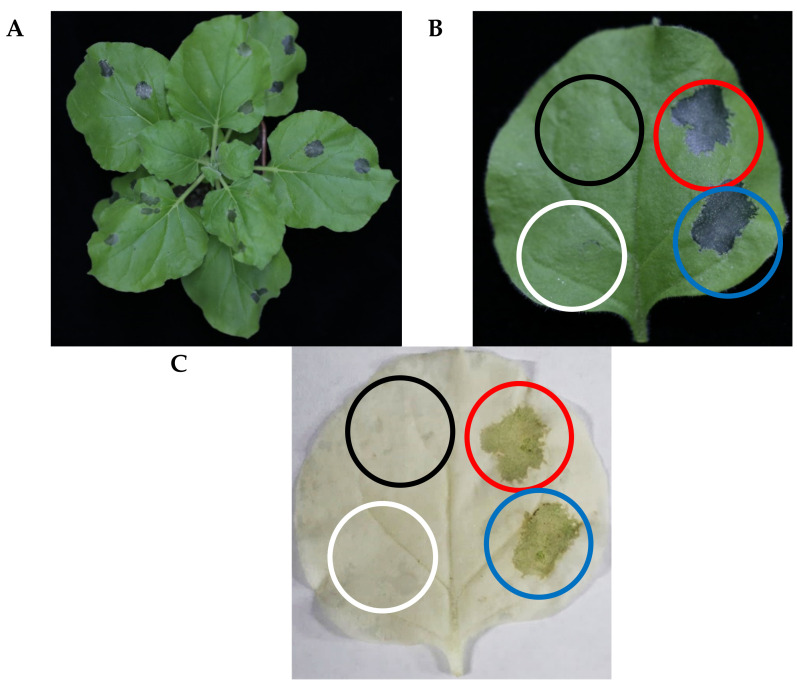
Hypersensitive response (HR) reaction induced by EPS isolated from *Clavibacter michiganensis* on leaves of *N. benthamiana.* Black, white, red, and blue circles indicate water treatment (mock control), 5 mg/mL sucrose treatment (negative control), 2 mg/mL EPS treatments, and 5 mg/mL EPS treatments, respectively. (**A**) Results were photographed 3 days after infiltration treatment for whole plants. (**B**) Single leaf of *N. benthamiana* (n = 6). (C) Decolorized leaf of *N. benthamiana* (n = 6).

**Figure 6 ijms-24-05324-f006:**
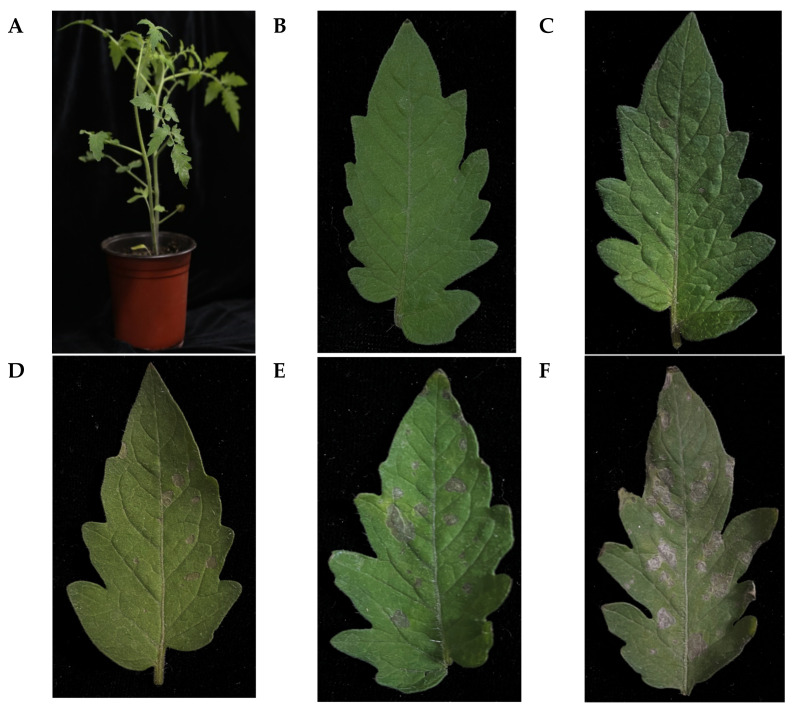
The pathogenicity test was mediated by the EPS isolated from *Clavibacter michiganensis* on tomato leaves. (**A**) Whole plants were photographed after 7 days of infiltration. (**B**–**F**) Tomato leaves were treated with water control and 0.5, 1, 2, and 5 mg/mL of EPS solution, respectively. All pictures were captured after 7 days of infiltration.

**Figure 7 ijms-24-05324-f007:**
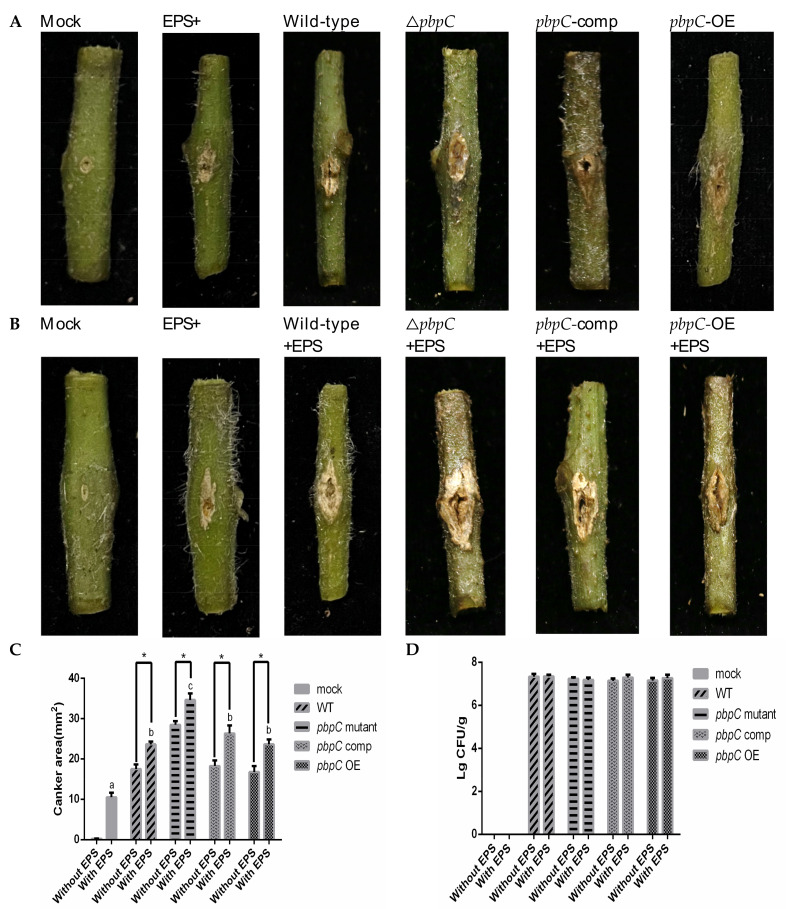
Results of the pathogenicity test of *Clavibacter michiganensis*, △*pbpC* mutant, *pbpC*-comp, and *pbpC*-OE strains with or without EPS on tomato seedlings. (**A**) Canker lesions on tomato seedling stems were caused by the pricked inoculation of bacterial cells and photographed at 14 days after inoculation (DAI). (**B**) Canker lesions on tomato seedling stems caused by the pricked inoculation of bacterial cells and EPS solution (treated after 1 day of pricked inoculation) and photographed at 14 DAI. (**C**) Canker area was calculated with Fiji software, and (**D**) bacterial cells were reisolated from stem segments. The mock means no *Clavibacter michiganensis* cells inoculated into tomato stem by pricked inoculation. Error bars represent standard error (n = 6). Different letters represent statistically significant differences by Duncan’s multiple range test (*p* < 0.05). Asterisks (*) indicate significant differences between inoculation with or without EPS (*p* < 0.05).

## Data Availability

All the data generated or analyzed in this study were included in this article and its Appendix A.

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
