# Peer review of "Deletion of pbpC Enhances Bacterial Pathogenicity on Tomato by Affecting Biofilm Formation, Exopolysaccharides Production, and Exoenzyme Activities in Clavibacter michiganensis"

_ijms, 2023, doi:10.3390/ijms24065324_

Round 1

Reviewer 1 Report

1. This paper provides a brief overview of the missing data on Clavibacter biofilm formation. The authors consider in biofilm, but other bacteria, in particular gram-negative ones, such as Burkholderia, Erwinia amylovora and Pseudomonas aeruginosa. I believe that this section should be supplemented with information on the biofilm formation of Gram-positive bacteria, in particular the Clavibacter genus. Information on this topic is available in the articles.

2. References in Introduction are outdated, only 17% of references are from 2020-2021. It should be noted that over the past 2 years, new interesting articles on bacterial biofilm formation and Clavibacter bacteria have appeared. Please read new works on the topic and refer to them.

3. Why, in the experiment, did the authors add tomato juice to the bacterial culture medium to obtain a culture, after interaction with the plant? What is it, what is this tomato juice? I believe that it would be more correct to introduce these plants into the liquid nutrient medium of tomatoes In vitro, after a period of colonization by the plant pathogen, to inoculate bacteria from tomato tissues and use these bacteria in the future for experiments.

4. The manuscript describes in great detail the method for determining the biofilm formation of bacteria. Is this your own method? If not, perhaps it is enough to give a link to the methodology and this will be enough.

5. Why was this tomato variety chosen? Is this variety resistant or susceptible to most pathogens?

6. In the section on statistical processing, it is indicated that Student's criterion is involved. It should be noted that this exchange criterion only for the normal distribution of data is extremely rare in biological experiments. Are you sure the primary data requires a normal return? The application criteria have been verified.

7. Fig. 3. (A) Images of a biofilm formed in polystyrene culture tubes, characterized by crystal violet 478 stained bands around the tube wall - is this a biofilm microscopy? The methodological part of the manuscript does not mention the detection of biofilm microscopy. Please explain.

8. The manuscript reads: “In this experiment, the EPS solution was diluted to a concentration of 1–5000 mg/l and sprayed onto fresh tomato leaves placed in a Petri dish with wet sterile gauze at 4°C.” I think that the authors have chosen too large a range of EPS concentrations. It is necessary to clarify the concentration of EPS used.

9. You write: "Exceptional frequency of EPS used to capture tobacco and tobacco (EPS from 0.1 to 5 mg/ml)". How reliable are the concentrations of EPS you use to the actual concentrations of EPS in the interaction of the host plant and the pathogen (for example, in accordance with the literature data)?

10. The phrase "In addition, no colonies of C. michiganensis were found when the plants were injected with EPS only." is not a discovery, it goes without saying.

11. In the Discussion, it is not enough to compare the obtained data with the published results on EPS and biofilm formation of bacteria family Clavibacter.

Author Response

Dear reviewer,

We have uploaded the response to the reviewer's comments and the revised manuscript with a cover letter. Please see the attachment.

Merry Christmas!

Thanks!

Reviewer 2 Report

It feels like the introduction section and the rest of the paper is written very differently. The results and the discussion sections are very difficult to understand because of the long sentences with very confusing/conflicting meanings. 

Following are some of the examples. However, it might be better if authors carefully read and edit results and discussion section.

Line 59 - “involves” -> involve

Line 73 - “polar” -> pole

Line 94 - “is able to”

Lines 122-123 - “As expected, only GS12102 and its derivatives were observed in qRT-PCR analysis to be without distinct changes in relative expression of pat-1 (Figure S2).”

Above sentence is confusing. Could you please explain why is it expected that pat-1 shows no change in WT, mutant, complemented, and OE strains? (Maybe I missed something)

Lines 132-133 - “significant six–eight and five–six-fold increases (Figure 1C and 1D), respectively. Transcription of pelA was also induced but to a lesser extent, as only a 3–4-fold increase was detected (Figure 1E).”

It might be better to follow the same format.

Lines 137-138 - “The relative expression levels of these two genes were did not change significantly fluctuate (Figure 1F and 1G)”

Please revise above sentence. 

Line 138-141 - could you please explain how chpC and ppaA participate in peptidoglycan biosynthesis and cross linking in Cm? If possible please provide reference as well. 

Line -163 - “violet” -> crystal violet

Lines 163-165 - “
Quantification by staining with violet and washing with ethanol revealed that biofilm formation by deletion mutants produced significantly (P < 0.05) higher with optical density at 590 nm (OD590) compared to the wild-type and pbpC-comp strains, both in BT0505 and GS12102”

Please revise this sentence.

Line 479 - please change “Date” to “data”

Line 495 - “The production of EPS isolated from wild type, â–³pbpC mutants, pbpC-comp and pbpC-OE strains of Clavibacter michiganensis strains “

Please revise this sentence.

Line 532 - “The canker lesions on tomato seedlings stems which were pricked inoculation with the mock control, mock with EPS, the wild type strain GS12102, the â–³pbpC mutant (â–³pbpC), the complementation strain of â–³pbpC (pbpC-comp) and pbpC over-expression strain (pbpC-OE) after 14 days.”

Please revise this sentence.

Line 510 - “The whole plant of EPS and negative control infiltrated were photographed after 7 days infiltration”

Please revise this sentence.

Lines 209-210 - Please revise this sentence.

Lines 215-221- Please revise this paragraph.

Lines 229-231 - Please revise this sentence.

Lines 232-233 - “Regarding cell morphology, the â–³pbpC mutants were not rod shape but instead had a thinner cell wall as observed by transmission electron microscopy and atomic force microscopy.”

Please provide reference for this statement?

Lines 233-234 -“On the other hand, the â–³pbpC mutants were more sensitive to lysozymes and had a higher stress response.” 

Please provide reference for this statement?

Line 257 - please change “stains” to “strains”

Lines 255-258 - The whole sentence is confusing. Please revise it.

Line 264-265 -“The composition and yields of EPS isolated from the infected plants by C. michiganensis were similar to EPS produced in vitro but differed among strains.”

Please revise this sentence.

Line 183 - “Effect of EPS isolation in virulence testing on tomato and hypersensitive reaction on tobacco.”
Please revise this sentence. It sounds like you are testing EPS isolation methods.

Line 271-273 - “
As a result, EPS presumably conglutinates together and blocks the plant stem xylem vascular resulting in a transportable moisture limit.”

I’m not sure if I understand this sentence. What is a “transportable moisture limit” in this sentence?

Lines - 291-292 -“As the cell wall peptidoglycan layer cross-links decreased, hypotheses regarding the interaction of C. michiganensis with tomato at peptidoglycan level require more details and researches.”

Please revise this sentence.

Lines 292-296 “Our original findings inform the current understanding of bacterial pathogenicity and highlight novel antimicrobial chemical controls that could be used to inhibit the biosynthesis of potential virulence factors. However, more details will be requisite to support the present hypothesis that the chemical composition of EPS contributes to plant–microbe interactions and the establishment of disease symptoms caused by phytopathogens, particularly Gram-positive bacteria.”

This paragraph is very confusing and difficult to understand. How do you use differences in pathogenicity related to peptidoglycan composition as an antimicrobial chemical control?

It’s not clear what you meant by “original findings”  and how it relate to current understanding and present hypothesis mentioned here.

No references are give for the whole paragraph.

Authors did not do any experiments related to composition of EPS in this study. Therefore, I’m not sure about the importance of that to this study.

Line 510 - “The whole plant of EPS and negative control infiltrated were photographed after 7 days infiltration.”

 Please revise this sentence.

Figure 6 - how many plants/leaves were used in this experiment (n)? Was this experiment repeated? Please include all these details.

Author Response

(The authors gave the same response as above.)

Reviewer 3 Report

This study generated a pbpC mutant in Clavibacter michiganensis and showed that this mutation caused several phenotypic alterations in this bacterium, including increased EPS production and exoenzyme secretion.  Although the study appears valuable and well designed, moderate editing is still needed.

The main limitations of this manuscript in its current form are largely in poor sentence structure and unclear communication in several passages where editing for clarity of English language is needed.  However, missing methods prevent full evaluation of the presented results.

In the methods section, for generation of the knockout mutant a reference is provided to Chen et al. 2021 for the methodology, but that paper does not provide sufficient details and in turn cites previous papers which give greater detail in the method (Kirchner et al 2001 and Lyu et al 2019).  I recommend that the papers providing the method details be referenced, either in place of or in addition to Chen et al.

Lines 95-100 are a complete oversimplification and contain inaccuracies.  It is unclear of what the authors mean by dehiscent stems, but I am not aware of any evidence of Erwinia amylovora causing dehiscence, actually circumstantial evidence that it may inhibit such processes (not implicated to be related to EPS).  To talk about EPS so generically is pretty unhelpful and to say that it has not been well studied seems inaccurate to me.  Since the authors try to use Erwinia as an example - this bacteria is known to produce 3 types of EPS (amylovoran, levan, and cellulose), each of which has known structure and characterized roles in disease - pathogenicity and virulence.  But that is far from the only phytopathogen with well characterized EPS.  The inaccurate generalizations must be corrected or removed throughout this paragraph.

Lines 232-234 - these are results (TEM, AFM, lysozyme comparison), not discussion items.  For these to be included here, they must be presented in the results section with any relevant data or images and accompanying methods must be included for evaluation.  The statements in these lines suggest that controlled experiments were conducted and there were statistically significant differences between wt and pbpC mutant cells.

The passage in lines 247-253 is so unclear in language that any meaning is completely obscured and these lines must be edited or removed.

Was any effort made to check the purified EPS for non-polysaccharide components to make sure the observed effects were not due to something that co-purified with the EPS?  Without this, caution should be used in making and discussing claims that the observed effects (such as HR) are due to EPS.

Throughout, several statements are made that should be supported with a reference citation.  Please check and add citations to support claims.

Author Response

(The authors gave the same response as above.)

Round 2

Reviewer 1 Report

I thank the authors for their work on the manuscript! Now she is much better. However, a little tweak is needed: 1. My remark 8 was answered by the authors. However, in the text of the manuscript, in the methodological part in the paragraph "Virulence and HR test", everything remained as before. It states that “In this experiment, the EPS solution was diluted to a concentration of 1–5000 mg/l and sprayed onto fresh tomato leaves placed in a Petri dish with wet sterile gauze at 4°C.” I ask the authors to make clarifications in the manuscript about EPS concentrations so as not to mislead the reader. 2. Statistical processing of data still raises doubts. The authors report that "The primary data were evaluated by mean with standard deviation and were compared using only t-test, with differences considered significant when p < 0.05." At the same time, the authors report that they did not check the distribution of the primary data for normality. I would like to note that t-test is relevant only with a normal distribution of primary data. This is very rare in biological experiments. Often the distribution is not normal, and then you need to use the nonparametric Mann-Whitney U-test. I believe that statistical data processing should correspond to the high level of the journal and should be carried out in accordance with the rules of statistics. I leave the final decision on this remark to the editor. 3. To the 9th question, the authors answered that the concentration of the introduced EPS corresponds to the titer of bacteria in plants 1010 CFU/mL. And what about in real conditions, is this bacterial titer achieved during pathogenesis in nature or did you deliberately choose a high infectious load on the plant? The answers to these questions should be included in the text. Happy New Year 2023 to the authors! I wish you success and new bright discoveries! Respectfully Yours, reviewer. December 29, 2022

Author Response

Dear reviewer,

We appreciate for the time and effort that you have dedicated to providing valuable feedback and insightful comments on our manuscript. Please see the response at the attachment.  We wish you Happy New Year 2023!

Sincerely

The author

Reviewer 2 Report

Based on the authors answers and the numerous errors still on the manuscript it looks like authors haven't done through review and professional editing.

Following are some examples:

Line 174- “Stains”

Line 94 - “when compared the non-biofilm-forming strains”

Lines 98-99 -

Lines 100-101 - “the bacterial exopolysaccharides (EPS) is implicated”

Line 112 - “symptom phenotypes”

Line 114- “In this study, we identified the mechanism of enhanced pathogenicity of the pbpC deletion mutant through increasing biofilm and EPS production and the activities of exoenzymes”

Line 132 - 134 “As expected, only strain GS12102 and its derivatives were observed in qRT-PCR analysis to beand without distinct changes in relative expression of pat-1, and the expression of pat-1 cannot be detected in strain BT0505 and its derivatives because of its lack of pCM2”

Lines 146-148 - “In addition, we considered serine protease as another pathogenic factor that has been reported comprehensively, thus, chpC and ppaA, which are located on the chromosome, were of interest for exploration of their potentially altered expression under a lack of pbpC in C. michiganensis”  

Author Response

(The authors gave the same response as above.)

Reviewer 3 Report

The manuscript is improved from the initial submission.  However, in your response to my prior comments, you have indicated that the results presented in the discussion lines 247-251 were from your prior study.  However, this manuscript claims that the mutants being analyzed in this study were generated for the current study.  Please clarify whether these are the same mutants being studied or new mutants.  If they are different, then a justification must be provided - if they are the same, this must be indicated in both the table of strains used as well as the methods.  Because you are dealing with pbpC mutants in different parental backgrounds, it is important to note for readers which strains the observations were made in (both in the current results and in reference to any prior results).

Further editing for English language is still needed.

Author Response

(The authors gave the same response as above.)

Round 3

Reviewer 1 Report

Dear authors!

My comments have been successfully corrected by You.

Thanks for the work done.

I wish You continued success!

Respectfully Yours, reviewer.

January 17, 2023

Author Response

Dear reviewer,

We truly appreciate for the time and the efforts that you have dedicated to providing specialized knowledge and valuable comments on our manuscript. We wish you make more progresses and have a brilliant future!

Respectfully Yours,

The authors.

Reviewer 3 Report

English language in manuscript is improved, and my concerns have been addressed.

Author Response

Dear reviewer,

We truly appreciate the time and effort that you have dedicated to providing valuable feedback and insightful comments on our manuscript. We have carefully checked the full text and made some minor revisions in “Introduction”, “Results” and “Discussion” section, all of which were in “track change mode” in this R3 version.

Thanks for your feedbacks!

Respectfully Yours,

The authors